# Improving Communication with Parents in the NICU during the COVID-19 Pandemic, a Study and Review of the Literature

**DOI:** 10.3390/children9111739

**Published:** 2022-11-11

**Authors:** Arieh Riskin, Shlomit Shlezinger, Lital Yonai, Frida Mor, Limor Partom, Elinor Monacis-Winkler, Keren Odler, Maria Goroshko, Ayala Gover

**Affiliations:** 1Department of Neonatology, Bnai Zion Medical Center, 47 Golomb Street, P.O. Box 4940, Haifa 31048, Israel; 2Rappaport Faculty of Medicine, Technion—Israel Institute of Technology, P.O. Box 4940, Haifa 32000, Israel; 3Department of Pediatrics, Bnai Zion Medical Center, 47 Golomb Street, P.O. Box 4940, Haifa 31048, Israel

**Keywords:** communication, parents, NICU, technology, satisfaction

## Abstract

Background: Communication with parents of sick premature and term infants in the NICU is complicated and challenging. Multiple efforts have been made to improve it, including the introduction of new electronic-based measures. Aim: We aimed to study the influence of implementation of a new communication technology on parents’ satisfaction with care in the NICU during the COVID-19 pandemic. Methods: Infants were video-recorded in their incubators or cots without being disturbed. These short films, with voice updates on the infant’s condition, were sent on a daily basis to their parents via a WhatsApp application. Results: Parents who chose to join the new communication project (study group) were older, and their infants were more premature. Parents were satisfied with this new communication modality. Satisfaction scores in both study and control groups were high, but not significantly different. Conclusions: Although the implementation of the new communication project was successful, we could not demonstrate significant improvement in satisfaction scores that were high in study and control groups, reflecting baseline high satisfaction. Further studies are needed employing other assessment tools in order to evaluate other aspects of parents’ satisfaction with new modalities of communication introduced to the NICU, and their effects on parents’ bonding with their infants.

## 1. Introduction

The admission of a sick and vulnerable infant to the neonatal intensive care unit (NICU) is an extremely stressful and complex experience for families [1]. It involves early separation from the newborn and disturbance to the bonding process, emotional distress and anxiety concerning the infant’s health and an unfamiliar overwhelming environment of critical care equipment, terminology and procedures. Over the years, modern neonatology has embraced the approach that parental involvement in care and parent–infant interactions are of the utmost importance to both the infant and parents, and may influence patient outcomes. Parents today need to be informed of their child’s condition, be involved in ethical and medical decision making and become true and active partners in caring for their baby.

Parent–staff communication in NICUs is often challenging due to the fragile emotional state of the parents, the interaction with multiple professional figures, the complexity of the infant’s medical condition and constraints related to the staff’s workload [2]. The staff is expected to understand and contain the anxieties and emotions of parents, and to support and comfort them through the most critical moments of their child’s illness [2,3,4]. Given the number and complexity of the inter-personal exchanges that take place in the NICU, the risk of misunderstanding, misinterpretation and conflict is high. Poor staff–family interactions not only reflect negatively on the baby’s care and are a source of distress and discontent for the parents, but are also a major cause of medico-legal litigation and increase the incidence of staff “burnout” [5]. Therefore, establishing good communication between parents and staff is essential in supporting the parents and optimizing neonatal care in NICUs. The search for ways to improve communication in order to keep parents updated, deeply involved in their infant’s care and alleviate their stress and anxiety is an ongoing effort.

The COVID-19 pandemic introduced further complexities in staff–parent communication, due to visitation restrictions, quarantines, physical distancing and other infection control measures [6]. We aimed to study the influence of implementation of a new communication technology on parents’ satisfaction with care in the NICU during the challenging period of the COVID-19 pandemic, and reviewed the literature on improving staff–parent communication.

## 2. Materials and Methods

### 2.1. Aim

This was a pilot study aiming to evaluate the feasibility and response to the use of electronic-based communication technology in our NICU.

### 2.2. Setting

The study took place in the NICU of the Bnai-Zion Medical Center in Haifa, at the time of COVID-19 pandemic, from 1 March 2020 to 30 June 2021. At this time, parental visits were limited to one parent at each visit, once a day; however, the length of the visit was not limited. No grandparents or siblings were allowed. When vaccinations became available, only vaccinated parents or unvaccinated parents with proof of a recent negative COVID-19 test were allowed to visit. During lockdowns, visits were authorized but could have been influenced by lack of public transportation for parents without access to a vehicle.

### 2.3. Participants

The participants comprised parents of preterm infants or term infants who were admitted to neonatal intensive care.

### 2.4. Intervention

Parents who gave their consent to the program received a WhatsApp message every morning, with a short video clip of their baby in the incubator or crib and an update from the nurse on their infant’s medical condition and “how his/her night had been” in terms of sleep, feeding, apneas, etc. Participation in the program was in addition to our routine practice of daily bedside updates by the staff, talks with the attending physician as needed and updating parents by phone upon any change or deterioration in their infant’s condition.

### 2.5. Ethics and Confidentiality

The study was approved by the Bnai-Zion Ethics Committee (IRB, BNZ-0096-19). In order to ensure confidentiality, the video clip with the voice message was recorded in a designated smartphone placed only in the NICU and available only to the NICU team. After sending the WhatsApp’s video clip or the recorded voice message to the parents’ cell phones, they were deleted from the NICU’s smartphone. However, since WhatsApp is a free public application, we could not fully guarantee that the video clip would not be exposed to others, due to illegal invasion or unintended exposure to the parents’ smartphones’ content when they were left unattended. Parents were thus informed about this possible, although uncommon, risk in the consent form.

### 2.6. Measures

Parents’ questionnaires included demographic information regarding the parents and the infants, as well as items related to parents’ assessment of their communication with medical staff, level of anxiety, and overall satisfaction. Items were rated by parents using a five-point Likert scale (ranging from 1—“do not agree at all” to 5—“strongly agree”). Questionnaires were distributed among parents in the NICU with and without the new communication program. Our questionnaire followed the one used by Globus et al. [7]. In addition, we tried to estimate parental stress and whether implementing the new communication tool could decrease it. For this, we adapted questions based on the concepts taken from the Parental Stressor Scale: Neonatal Intensive Care Unit (PSS: NICU) questionnaire [8]. We compared the questionnaires of parents who participated and did not participate in the new communication project. Although we originally planned on a gradual introduction of the project with comparison of pre- and post-implementation-era questionnaires, the COVID-19 pandemic, lockdowns and limitations on parents’ visits to the NICU dictated rapid immediate implementation of the new electronic communication option for parents who desired it during those rough times.

### 2.7. Statistical Analysis

Data were statistically analyzed using SigmaPlot, version 11.0 (Systat Software Inc., San Jose, CA, USA) and Minitab^®^, version 16.2.2 (Minitab Inc., State College, PA, USA and Coventry, UK) using descriptive statistics, *t*-test for comparing continuous variables and chi-square for comparison of categorical data. The corresponding non-parametric tests were employed (Mann–Whitney Rank Sum test on medians) if the distribution of the results was not normal.

## 3. Results

During the year and a half study period (1 March 2020 to 30 June 2021), we collected 45 questionnaires from parents who participated in the new electronic communication project (study group) and 51 questionnaires from parents who received an offer but chose not to participate in this project (control group). Their characteristics are presented in Table 1.

Parents in the study group were significantly older (by an average of three years). Most parents were married or lived as couples. There were more parents to preterm infants (born at fewer than 30 weeks of gestation) in the intervention group (6 vs. 1), and most of the parents of term infants admitted to the NICU were in the control group (31 vs. 11). Most of the infants with low birth-weights were in the intervention group according to gestational age distribution, while most of the infants with a birth-weight above 2 kg were in the control group. There were no differences in the number of siblings. There were no differences in the frequency of NICU visits or the time of visits by the parents in the study and control group, yet parents in the study group tended to schedule their visits to the NICU more in the morning and noon hours. Parents in the control group tended to call the NICU more frequently (2.7 vs. 2.2 calls per day on the average).

There were no significant differences in satisfaction scores between the study and control groups (Table 2). Stress and anxiety scores were low in both groups, although they were slightly higher in the intervention group. Parents in both groups graded medical treatment as the most important domain of care, but this issue was significantly more important to parents in the study group (p = 0.013). Although issues related to privacy were not considered of high importance for both groups, parents in the control group were significantly more concerned with it (*p* = 0.040). Overall, satisfaction with treatment and staff attitude during hospitalization was very high and close to the maximum of 10 out of 10 in both groups.

Satisfaction with the new communication project in the study group was very high (average score of 9.6 ± 1.0 out of 10 with a median of 10, range 6–10 and interquartile range of 10, 10).

## 4. Discussion

### 4.1. Parent–Staff Communication in the NICU

The psycho-relational problems in neonatal intensive care units (NICUs) have only recently been properly addressed [5]. In the past, parents had little or no access to NICUs and therefore insufficient contact with their babies and the medical staff, resulting in limited opportunities for communication. This has changed profoundly over the last two decades, and NICUs are now accessible to parents [9], allowing opportunities for enhanced parent–infant interactions and parent–staff communication, which is now considered an essential element of family-centered care [10,11,12]. The issue of communication with parents in the NICU is complex and multifaceted [4,13]. Some specific factors make communication in the NICU particularly problematic: the baby’s clinical condition, the emotional and workload conditions of the medical staff, the emotional state of the parents, the critical care environment and the interaction of multiple healthcare providers with the parents [5]. Frequent and sensitive communication from neonatal staff is important to alleviate parental stress and to ensure that parents understand their baby’s condition. It also empowers and involves parents in their baby’s care. Unfortunately, time constraints often limit the staff’s ability to fully address parent’s needs. A lack of regular and informative communication from neonatal staff is a common reason for parental complaint [14]. Parents feel that important information is not relayed satisfactorily almost 50% of the time (i.e., either too much, too little, or not explained at all) [15]. Many times, the parents, searching for information that is more detailed over the web, report disappointing experiences. Therefore, specific training of the staff in communication is fundamental for optimal results, as well as implementation of modern technology to enhance the quality of staff–parent communication [2,3,11,12,15,16,17,18,19].

### 4.2. Communication-Targeted Simple Interventions

Relatively simple communication-targeted interventions, such as educating providers, distributing contact cards to families and displaying a poster of providers in the NICU, can improve the quality and the quantity of parent–provider communication and increase parent satisfaction [20]. Individual baby diaries are also a relatively simple, practical and cost-effective tool that enhances communication between parents and staff in the NICU [14]. Family-centered rounds with parent participation are encouraged by studies; however, many infants have long hospitalizations with parents being unable to attend rounds. In a small study of audio recordings of NICU rounds, it was shown that both providers and parents could agree to recording rounds and that these recordings can be reliably analyzed. Unfortunately, the study found that the proportion of psychosocial and emotional communication during NICU rounds was negligible, and families in the intervention group were less satisfied with communication. The authors concluded that families who are primed to expect better communication, such as those participating in a communication intervention, may be less satisfied with standard care, and this should be taken into account when considering interventions to improve communication with parents in the NICU, improve their satisfaction or decrease their stress [21].

### 4.3. Advanced Technological Solutions

Technological solutions for conveying information to parents are increasingly explored. Patient-centric web methods are being developed to leverage technology for mapping information from the electronic patient record and pushing relevant professional content using internet and cell phone technologies. This can become an important informatics resource, complementing and enhancing face-to-face communication through personalization of education and advice to the parents [22]. However, the personalization of information over multimedia databases involves multiple steps of extracting the reference literature, summarizing it and matching it to a query based on contexts of extraction from clinical databases, a project that can require a considerable period to complete. Many issues of text mining, semantic information modeling and query definition and refinement are involved and present considerable challenges. NICU-specific customization of such a system of information in a patient-centered manner that will address issues unique to NICU care context for parents would also need to consider inter-cultural and language issues [22].

Until such sophisticated systems are developed and implemented, simpler and more readily available internet- and cell-phone-based solutions can be used to enhance communication of the medical team in the NICU with the parents on a daily basis. Studies have demonstrated improved family satisfaction and readiness for discharge with a daily electronic medical record (EMR)-derived report with laymen terminology distributed to families unable to participate in daily rounds [23]. Implementation of an EMR-generated daily patient update letter is feasible, and may improve communication. It can also improve at least one aspect of parent engagement—perceived competence to manage information in the NICU [24,25]. Another such project is the BUDS project launched by the neonatal research team at the Chelsea and Westminster Hospital in October 2016. The BUDS project developed a communication tool that directly extracts infant information from the neonatal EMR and provides it to parents electronically through a mobile phone application in order to improve communication with parents and increase their satisfaction [25].

Another even simpler and readily available short-message services (SMS)-based communication technique was successfully implemented to provide daily updates to the parents regarding the health status of their preterm infant in the NICU. It was shown that the SMS updating system is an easy and user-friendly technology that enriches the modalities of information delivery to parents of hospitalized preterm infants. It was also found to be a complementary and useful tool for encouraging and improving personal communication between parents and medical staff in the NICU [7]. In a survey that assessed mobile use and communication preferences in a population of NICU mothers, it was found that the mothers viewed receiving electronic messages about their babies favorably, and text messaging was the preferred platform. Although the majority of mothers felt electronic messaging could improve provider–parent communication and reduce parental stress, they stressed that it should not replace verbal communication [26].

Elevated stress in parents of infants hospitalized in the NICU is noted to exacerbate fear and uncertainty. Parents who were more educated had higher expectations and more knowledge of illnesses and the medical treatment being given to their child. Therefore, they had a better idea of their child’s prognosis when looking at their infant’s appearance and behavior [27]. This finding stresses the importance of the use of readily available electronic communication applications that provide parents with an on-line visual of their infant’s current appearance and state. There are several reports of the successful use of such applications to enhance communication with parents of preemies in the NICU, especially at times when parents are not able to be by their baby’s cot-side—a situation considered to be an additional source of stress. Since the implementation of such a system, video updates were made available for all babies. vCreate, a secure video messaging platform adopted by the Royal Hospital for Children in Glasgow, the largest pediatric teaching hospital in Scotland, is such a system. Since the implementation of this system, video updates have been made available for all babies. vCreate enables nurses to record video messages securely and send them to parents. The parents can access the clips at any time and through any device. Then, prior to the baby returning home, parents can download the clips and save them for later use in their baby care diary. vCreate was developed in close collaboration with the hospital’s governance and IT team to ensure that the solution reflected data protection and security criteria. In accordance with data protection regulations, once the parents of a baby have downloaded their clips, the administrator is alerted and reminded to delete the clips. Parents say that receiving these video messages of their baby provides a sense of reassurance [28]. Another such solution is the use of Skype and an iPad, a technique employed in the NICU of the University of Virginia medical center, which allows parents who cannot be in the NICU to be in real-time contact with doctors, nurses and most importantly their babies [29]. Similarly, a cell phone app named EASE developed for the NICU of Orlando Hospital in Florida allowed NICU parents to be in close on-line contact even when hurricane Irma hit Florida and the hospital was put on lockdown [30].

Recently, applications such as the Angel Eye mobile app, developed by Angel Eye Camera Systems, have been developed, offering hospitals a secure, internet-based, video and audio solution allowing family and friends to interact with hospitalized newborns at any time. These applications are readily available and compatible for use with iPhones, iPads and iPods [31].

The literature on the use of videoconferencing, videophone and commercially available modalities [32] mainly evaluated parents’ perception of technology use, healthcare providers’ perceptions of technology use and objective outcomes, such as parental anxiety or stress or infant length of stay. Overall, parents and healthcare providers perceived the various interventions quite favorably. Only a minority of parents felt guilty for not being able to be with their infant when they viewed them on a camera [33]. However, no more than a few significant differences were found for objective measures across studies. This calls for further rigorous research with diverse samples, as research in this area is quite limited so far [32].

### 4.4. Staff–Parent Communication in the NICU during the COVID-19 Pandemic

The dramatic onset of the COVID-19 global pandemic caused hospital administrators and NICU leadership worldwide to change visitation policies as per public health guidelines, a necessary measure due to the state of emergency. Some centers restricted all parent visits, and some allowed only one parent to visit their child in the NICU. Visitation opportunities were further affected by isolations and quarantines. While intended to protect families and healthcare providers from illness, these policies significantly limited parental presence and involvement in their child’s care [34]. Bembich et al. interviewed a small number of parents whose infants were hospitalized at the time when visitation restrictions and a shift from a 24/7 open-ward policy to restricted visits began. They report that NICU visitor restrictions had a dysphoric and challenging psychological impact on parental experience, and accentuated their emotional distress and suffering [6]. Another study found that access to technology such as texting and video chat was positively associated with higher mental health scores [35].

Campbell-Yeo et al. conducted a survey to explore the use and impact of technology in 38 NICUs across Canada in the period of visitation restrictions during the COVID-19 pandemic [36]. They found that two-thirds of parents had primarily communicated with the staff via traditional telephone calls, and video calls and text messaging were less often used. Some parents reported no communication at all with the NICU care team. The majority of parents reported being comfortable with technology use; however, some reported having difficulty hearing the care team or connecting virtually. Video calling through platforms such as FaceTime was beneficial in helping parents feel closer and connected; however, there were concerns about overstimulating the infant. In NICUs where secure webcam systems were available, families expressed that these systems helped them feel closer to the infant. Restrictive use of technology was reported as a source of stress for families.

In one unit in Italy [37], NICU parents were updated through a daily video call via smartphones and received information from the physicians about the clinical course of their babies in the presence of a psychologist able to provide support to the families. Parents could see their babies during the video calls, and these were made when the infants were sleeping or feeding, or during procedures. Parents could also receive telephone updates from physicians for approximately 15 min at designated times to clarify any doubts about the clinical status of their newborns. The authors found that parents were overall satisfied with the information received, consideration of their privacy and the level of intervention-reduced stress. However, their results were not as good as those from a previous cohort receiving family-centered care prior to COVID-19, because naturally, video calls cannot replace physical contact. Another study implemented a virtual communication project which included virtual rounds and communication with the NICU team using an iPad, and reported an overall positive experience, despite technical challenges and slow devices [38].

### 4.5. Our Parent Communication Project

In our study, we found that satisfaction with our new electronic-based communication project during the COVID-19 pandemic was very high, but we could not demonstrate a difference as compared to the control group, because satisfaction with care was very high in both groups.

However, our study design allowed for selection bias; parents who chose to be in the intervention group were older and less concerned with the issue of privacy. Their infants were more preterm with a lower birth-weight, so they were expected to have longer hospitalization in the NICU. We assume this could have been one of the reasons that led them to agree to participate. Another selection bias could have been related to parents’ anxiety levels, which were slightly higher in our intervention group; this could be explained by the desire of those who were more anxious to see their infants on the video. In fact, our results may indicate that we were able to maintain a high level of satisfaction with the use of our intervention, despite the worse initial clinical setting in the intervention group. There were no other significant differences in the characteristics of the parents in the study and control groups. We did not examine parent–infant attachment or the parental sense of personal efficiency, and this is another limitation of this study.

As expected, parents in the control group tended to call the NICU more times per day, although the difference was not statistically significant. It is possible that daily WhatsApp communication slightly alleviated parents’ stress and anxiety in the study group, and made them call the NICU on the phone less frequently to ask how their infant was doing.

While conceptualizing this new communication project, we were concerned that it might create high expectations regarding communication with the parents and that the team may not be able to fulfill them on occasion, particularly when there is an increased or urgent workload or a paucity of team members. Our results of a very high level of satisfaction disputed such concerns.

Satisfaction with our care and attitude were also reflected in comments that parents wrote on the questionnaires. In agreement with previous studies on technology-based communication, it seems that this project answered a true need of parents to feel more involved with and connected to their infants, and provided both parents and medical teams with an additional important communication tool. It definitely did not lower their satisfaction with care.

In retrospect, the questionnaire that we adopted based on a previous study on communication with parents in the NICU [7] may have been less suitable for our study and was not sensitive enough to expose and identify differences between our study and control groups. This was most probably due to very high satisfaction scores on all the items regarding care and attitude in our NICU to start with, as seen in our control group. Another possibility is that our results were influenced by the Hawthorne effect on our staff, as when participants are aware of their participation, the studied behavior could change. It is possible that our engagement with issues of communication in the NICU and the introduction of the new technology to improve it raised our team’s awareness of the importance of communication with and attitude toward the parents and thus contributed to an improvement in parents’ satisfaction both in the study and control groups.

Further studies are needed in other NICUs using different questionnaires in order to demonstrate improvements in parents’ experience and satisfaction resulting from adoption of this new technology-based communication method. In addition, it could be of great interest to investigate in future studies how such electronic-based communication methods affect how parents bond with their infants.

## 5. Conclusions

Communication with parents of sick premature and term infants in the NICU is complicated and challenging. Multiple efforts are made to improve it, including the introduction of new electronic-based measures. The COVID-19 pandemic introduced further complexities in staff–parent communication, due to visitation restrictions, quarantines, physical distancing and other infection control measures. We studied the influence of the implementation of a new communication technology on parents’ satisfaction with care in the NICU during the COVID-19 pandemic. Although the implementation of the new communication project was successful, we could not demonstrate a significant improvement in satisfaction scores, which were high in the study and control groups, reflecting baseline high satisfaction. Further studies are needed employing other assessment tools in order to evaluate other aspects of parents’ satisfaction with new modalities of communication introduced to the NICU, and their effects on how parents bond with their infants.

## Figures and Tables

**Table 1 children-09-01739-t001:** Parent and infant characteristics in the study and control groups (with and without the new electronic communication).

	No Intervention *n* = 51	With Intervention *n* = 45	*p*
Age	31.9 ± 5.6	34.9 ± 6.0	
	33.0 (26.5, 36)	35.0 (30, 38.5)	0.017 ^¶^
Married/Couple	92.1%	91.1%	1.000 *
Education years	14.7 ± 3.0	14.4 ± 3.3	0.052
	15.0 (12, 16)	14.0 (12, 16.25)	
Infant’s gestational age < 30 weeks	2.0%	13.3%	0.853 *
Infant’s gestational age ≥ 37 weeks	63.3%	24.4%	<0.001 *
Infant’s birth-weight < 1250 g	6.8%	18.2%	0.197 *
Infant’s birth-weight ≥ 2000 g	84.1%	47.7%	<0.001 *
Number of siblings	2.4 ± 2.2	1.6 ± 0.7	0.210
	2.0 (1, 3)	1.0 (1, 2)	
I generally visited the NICU at least once a day	96.0%	93.2%	0.662 **
Average hours in NICU per day	5.7 ± 3.7	5.3 ± 2.8	0.907
	5.0 (3, 7)	5.0 (3, 8)	
Time of daily visit to the NICU:			0.103 *
- All day long/non specific	79.6%	62.2%	
- Morning to noon	20.4%	37.8%	
Average number of calls to the NICU per day	2.7 ± 1.5	2.2 ± 1.3	0.077
	2.0 (1, 4)	2.0 (1, 3)	

Data presented as mean ± standard deviation in the first line, followed by median (interquartile range 25–75%) in the second line or percent (%) for categorical data. All comparisons analyzed using Mann–Whitney Rank Sum Test on medians, unless mentioned otherwise. * Chi-square. ** Fisher Exact Test. ^¶^
*t*-test.

**Table 2 children-09-01739-t002:** Parents’ satisfaction questionnaire scores in the study and control groups.

	No intervention n = 51	With intervention n = 45	*p*
The physician was available when needed ^a^	4.7 ± 1.2 5.0 (4, 5)	4.7 ± 0.6 5.0 (5, 5)	0.620
The nurse was available when needed ^a^	5.0 ± 0.2 5.0 (5, 5)	4.9 ± 0.2 5.0 (5, 5)	0.554
The physician was patient in answering my questions ^a^	4.9 ± 0.3 5.0 (5, 5)	5.0 ± 0.7 5.0 (5, 5)	0.830
The nurses were patient in answering my questions ^a^	4.9 ± 0.4 5.0 (5, 5)	4.9 ± 0.3 5.0 (5, 5)	0.853
I felt comfortable approaching the physicians ^a^	4.7 ± 0.6 5.0 (4.25, 5)	4.8 ± 0.7 5.0 (5, 5)	0.156
I felt comfortable approaching the nurses ^a^	4.8 ± 0.4 5.0 (5, 5)	4.9 ± 0.4 5.0 (5, 5)	0.188
I received clear and understandable information from the physicians about my infant’s medical status ^a^	4.9 ± 0.4 5.0 (5, 5)	4.8 ± 0.6 5.0 (5, 5)	1.000
I received clear and understandable information from the nurses regarding my infant’s medical status ^a^	4.8 ± 0.5 5.0 (5, 5)	4.9 ± 0.4 5.0 (5, 5)	0.642
I regularly received information regarding my infant’s medical status ^a^	4.7 ± 0.6 5.0 (5, 5)	4.8 ± 0.6 5.0 (5, 5)	0.337
I trusted the physicians; I felt we were in good hands ^a^	4.9 ± 0.3 5.0 (5, 5)	4.9 ± 0.3 5.0 (5, 5)	0.860
I trusted the nurses; I felt we were in good hands ^a^	4.9 ± 0.5 5.0 (5, 5)	4.9 ± 0.2 5.0 (5, 5)	0.813
I felt I was involved in the care of my infant during the hospitalization ^a^	4.8 ± 0.6 5.0 (5, 5)	4.7 ± 0.6 5.0 (5, 5)	0.921
I was updated regarding my infant’s examination results ^a^	4.9 ± 0.7 5.0 (5, 5)	4.6 ± 0.7 5.0 (4, 5)	0.140
I was pleased with the frequency with which I received information regarding my infant ^a^	4.8 ± 0.5 5.0 (5, 5)	4.7 ± 0.6 5.0 (4, 5)	0.250
I was pleased with the manner in which I received information about my infant ^a^	4.8 ± 0.5 5.0 (5, 5)	4.8 ± 0.5 5.0 (5, 5)	0.991
Importance of domains of care in NICU (graded 1–5) ^b^			
- Parents’ instructions	3.5 ± 1.3 3.0 (2.75, 5)	3.1 ± 1.2 3.0 (3, 4)	0.211
- Medical treatment	1.5 ± 1.2 1.0 (1, 1)	1.0 ± 0.2 1.0 (1, 1)	0.013
- Parents’ facilities in the NICU	3.9 ± 1.0 4.0 (3, 5)	3.6 ± 1.1 4.0 (3, 4)	0.261
- Privacy	3.8 ± 1.1 4.0 (3, 5)	4.2 ± 1.0 5.0 (3.5, 5)	0.040
- Updates about infant’s condition	2.3 ± 0.9 2.0 (2, 3)	2.0 ± 0.7 2.0 (2, 2)	0.182
How anxious I feel now? (marked on a scale of 0 (not anxious) to 10 (very anxious)) ^c^	1.6 ± 2.5 0 (0, 2.75)	2.5 ± 3.1 1.0 (0, 5)	0.154
How anxious I feel when I think about taking my infant home? (marked on a scale of 0 to 10) ^c^	1.9 ± 2.5 1.0 (0, 4)	2.3 ± 2.6 1.0 (0, 4)	0.415
Overall level satisfaction with treatment and staff attitude during hospitalization (marked on a scale of 0 (unsatisfied) to 10 (very satisfied)) ^c^	9.6 ± 1.4 10.0 (10, 10)	9.7 ± 0.6 10.0 (10, 10)	0.814

^a^ Questions 1–15 were graded on a Likert scale from 1 (never) to 5 (always). ^b^ The parents were presented with a list of domains of care in the NICU and graded them based on their importance between 1 (most important) to 5 (least important). ^c^ Anxiety and satisfaction were marked on a graphic scale from 0 (lowest) to 10 (highest). Data are presented as mean ± standard deviation in the first line, followed by median (interquartile range 25–75%) in the second line. All comparisons analyzed using Mann–Whitney Rank Sum Test on medians, unless mentioned otherwise.

## Data Availability

The data presented in this study are available on request from the corresponding author.

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
