# Peer review of "Improving Communication with Parents in the NICU during the COVID-19 Pandemic, a Study and Review of the Literature"

_children, 2022, doi:10.3390/children9111739_

Round 1

Reviewer 1 Report

Thank you for the opportunity to review this interesting article “Improving communication with parents in the NICU during the 2 COVID-19 pandemic, a study and review of the literature.“

The article was very well written. There are no concerns regarding grammar or language.

I have the following comments:

1.     The aim of the study should be put at the end of the introduction

2.     Please restructure the Material and method section

3.     Explain please “lockdowns and limitations on parents' visits” in your country; because depending on the restriction, participating parents may have been coming as much as those who were not in the study. Was this visit time collected as a comparison variable?

4.     Were there more complications or bad news in one group?

5.     The tool is only used to give standard or simple news without emotional impact on the parents?

6.     Why have studied only the stress which seems to be logical but not in communication strategy because the emotions are guided by the words used and therefore very operator dependent finally. I would have appreciated to have a follow-up on attachment or on the feeling of personal efficiency (like "I am a good mom or dad").

Author Response

We thank the reviewers for their helpful comments.

Reviewer 1:

  1. The aim of the study should be put at the end of the introduction

We have reconstructed the end of the introduction and have stated the aim of the study.

  1. Please restructure the Material and method section

We restructured the section into numbered subsections. We hope this improves presentation of the information.

  1. Explain please “lockdowns and limitations on parents' visits” in your country; because depending on the restriction, participating parents may have been coming as much as those who were not in the study. Was this visit time collected as a comparison variable?

Thank you for this important comment. Our visiting limitations were added in detail to the setting description of the Materials and Methods section. Visits times were comparable between the groups, please see Table 1.

  1. Were there more complications or bad news in one group?

Unfortunately, we have not collected this data, however since the intervention group included more preterm infants with lower birth weights, it is possible that their hospital course was more complicated. We feel that this may emphasize the effectiveness of the intervention which allowed for the same level of parent satisfaction in the more complicated group as in the less complicated one.

  1. The tool is only used to give standard or simple news without emotional impact on the parents?

The tool was indeed used to give simple news and was meant to give the parents an additional way of feeling connected to their infant.

  1. Why have studied only the stress which seems to be logical but not in communication strategy because the emotions are guided by the words used and therefore very operator dependent finally. I would have appreciated to have a follow-up on attachment or on the feeling of personal efficiency (like "I am a good mom or dad")

Thank you for this insightful comment. We agree the tool is operator dependent, although naturally all verbal communication strategies are somewhat operator dependent. We chose to evaluate parental satisfaction with care, stress and anxiety because these were common measures examined in previous literature. We regret not having done a follow up on attachment or on personal efficiency, as this would have been appropriate, as you suggested. We have added this to the discussion as a limitation of the study.

Reviewer 2 Report

The authors have submitted a manuscript outlining their experience with a novel communication method during COVID with families of patients in the NICU. I have a few minor questions.

1) How did the authors handle families who needed translation services?

2) I didn't see if the families were able to respond and interact or if it was just a video update and they needed to respond in a different format.

3) Certainly there was some bias introduced with the lack of randomization, do the authors suspect that if the populations had been randomized, that they would have found a difference?

Author Response

We thank the reviewers for their helpful comments.

Reviewer 2:

  1. How did the authors handle families who needed translation services?

Our nursing staff is linguistically diverse and fluent in all common languages in our country – Hebrew, Arabic and Russian, however most parents were comfortable with video messages in Hebrew. When preferred, messages were recorded in the parents' mother tongue. We have full translation services available in our hospital but they were not required for this study.

  1. I didn't see if the families were able to respond and interact or if it was just a video update and they needed to respond in a different format

Our study offered only a video update and not a live interaction. Parents were able to respond to the update and the staff via our regular communication routes – phone calls or in person.

  1. Certainly there was some bias introduced with the lack of randomization, do the authors suspect that if the populations had been randomized, that they would have found a difference?

Our study was not randomized as we felt it would be unethical to limit participation of any parent who was interested in this form of communication at this unique time of COVID-19 restrictions. We do speculate that if we had randomized the groups, we would have found a difference, because we received such warm compliments and appreciation in textual comments from the parents in the intervention group.

Round 2

Reviewer 1 Report

Thank you for correction and to have discussed the limits of this study.